# The matricellular protein CCN5 induces apoptosis in myofibroblasts through SMAD7-mediated inhibition of NFκB

Mai Tuyet Nguyen[1], Min-Ah Lee[1], Young-Kook Kim[2], Hyun Kook[3], Dongtak Jeong[4], Seung Pil Jang[5], Tae Hwan Kwak[5], Woo Jin Park[1]*

1 College of Life Sciences, Gwangju Institute of Science and Technology, Gwangju, Republic of Korea,
2 Department of Biochemistry, Chonnam National University Medical School, Hwasun, Jeollanam-do, Republic of Korea, 3 Department of Pharmacology, Chonnam National University Medical School, Hwasun, Jeollanam-do, Republic of Korea, 4 Department of Molecular & Life Science, College of Science and Convergence Technology, Hanyang University-ERICA, Ansan, Gyeonggi-do, Republic of Korea,
5 BethphaGen, S3-203, Gwangju Institute of Science and Technology, Gwangju, Republic of Korea

* woojinpark@icloud.com

**Data Availability Statement:** All relevant data are within the manuscript and its Supporting Information files

## Abstract

We previously showed that the matricellular protein CCN5 reverses established cardiac fibrosis (CF) through inducing apoptosis in myofibroblasts (MyoFBs) but not in cardiomyocytes or fibroblasts (FBs). In this study, we set out to elucidate the molecular mechanisms underlying CCN5-mediated selective apoptosis of MyoFBs. We first observed that the apoptotic protein p53 and the anti-apoptotic protein NFκB are simultaneously induced in MyoFBs. When the expression level of p53 was suppressed using a siRNA, CCN5 did not induce apoptosis in MyoFBs. By contrast, when NFκB signaling was inhibited using IKK VII, an IκB inhibitor, MyoFBs underwent apoptosis even in the absence of CCN5. SMAD7 is one of the downstream targets of CCN5 and it was previously shown to potentiate apoptosis in epithelial cells through inhibition of NFκB. In accordance with these reports, when the expression of SMAD7 was suppressed using a siRNA, NFκB signaling was enhanced, and CCN5 did not induce apoptosis. Lastly, we used a luciferase reporter construct to show that CCN5 positively regulated SMAD7 expression at the transcriptional level. Collectively, our data suggest that a delicate balance between the two mutually antagonistic proteins p53 and NFκB is maintained for MyoFBs to survive, and CCN5 tips the balance in favor of the apoptotic protein p53. This study provides insight into the anti-fibrotic activity of CCN5 during the regression of CF.

## Introduction

Heart failure (HF) remains a leading cause of mortality and morbidity worldwide. It is a chronic disease that is associated with adverse cardiac outcome [1, 2]. HF is often initiated after myocardial injuries that are caused by a variety of events, including myocardial infarction, hypertension, heart valve disease, and myocarditis, and is usually accompanied by

**Funding:** W.J.P. was supported by grants from National Research Foundation of Korea (2022R1A2C1004256) funded by the Korean Government (MSIP). Y.-K.K. and H. K. were supported by grants from National Research Foundation of Korea (2018R1A2B3001503, 2022R1A4A200076711, 2021R1A2B5B0200150). D.J. was supported by grants from National Research Foundation of Korea (2021R1A2C1008058, 2021R1A4A5032463).

**Competing interests:** We state T.H.K. and W.J.P. have co-ownership interest in BethphaGen. The other authors have declared that no conflict of interest exists. There are no patents, products in development or marketed products associated with this research to declare. This does not alter our adherence to PLOS ONE policies on sharing data and materials.

**Abbreviations:** CCN5 (WISP2), WNT1 Inducible Signaling pathway Protein 2; HF, Heart failure; FBs, Cardiac fibroblasts; ECM, Extracellular matrix; MyoFBs, Myofibroblasts; TGF-β, Transforming Growth Factor-β; AngII, Angiotensin II; CM-Con, control conditioned media; CM-CCN5, CCN5-containing conditioned media.

inflammation and fibrosis [1–3]. Cardiac fibroblasts (FBs) play an important role in maintaining the function of cardiomyocytes through regulation of turnover of the extracellular matrix (ECM) proteins and communication with cardiomyocytes [4–6]. In addition to these supportive roles, FBs play a primary role in the progression of HF. Upon pathological stimuli, FBs undergo proliferation and trans-differentiation to myofibroblasts (MyoFBs) [7–9]. MyoFBs secret excessive amounts of ECM proteins, which are responsible for the increased ventricular stiffness and diastolic dysfunction observed in HF patients. Therefore, preventing FB trans-differentiation and/or reversing the differentiation of MyoFBs could be efficient strategies to treat HF. Effective anti-fibrotic therapies are currently unavailable.

The CCN family (CCN1-6) of matricellular proteins are associated with diverse cellular processes including fibrosis, angiogenesis, cell differentiation, and wound repair [10, 11]. Our group previously showed that CCN5, also known as WNT1-inducible signaling pathway protein 2 (WISP-2), inhibits cardiac fibrosis (CF) through inhibition of endothelial–mesenchymal transition and trans-differentiation of FBs to MyoFBs [12–16]. More strikingly, CCN5 reverses pre-established CF through induction of apoptosis in MyoFBs but not in cardiomyocytes or FBs [14]. However, the molecular mechanisms underlying the selective pro-apoptotic activity of CCN5 are largely unknown.

In this study, we attempted to further elucidate the role of CCN5 in MyoFB-specific apoptosis. FBs were freshly isolated from rat hearts, and MyoFBs were obtained by trans-differentiating by treatment with TGF-β. Our results suggest that a balance between pro-apoptotic p53 and anti-apoptotic NFκB is established in MyoFBs and that CCN5 tips over the balance to favor p53, which results in apoptosis through SMAD7-mediated inhibition of NFκB. This work provides insight into the role CCN5 in reversing CF and a basis for the development of anti-fibrotic therapies.

## Materials and methods

### Ethics statement

Animal experiments using Sprague-Dawley rats were granted by approval of the Institutional Animal Care and Use Committee (IACUC) of Gwangju Institute of Sciences and Technology (GIST-2020-056). The investigation conforms to the Guide for the Care and Use of Laboratory Anials published by the US National Institutes of Health (NIH Publication No. 85–23, revised 1996). Primary cardiac fibroblasts isolation procedures were performed under inhalational anesthesia with isoflurane gas (N2O:O2/70%:30%), and all efforts were made to minimize suffering. The eight-week-old male Sprague-Dawley rats were obtained from Daehan Biolink (Korea) and used for all isolation experiments.

### Cell culture

Adult cardiac FBs were isolated from the hearts of male 300–350 gram Sprague Dawley rats. The heart was quickly removed from the chest, and the aorta was retrogradely perfused at 37˚C for 3 min with calcium Tyrode buffer (137 mM NaCl, 5.4 mM KCl, 1 mM MgCl$_2$, 10 mM glucose, 10 mM HEPES pH 7.4, 10 mM 2, 3-butanedion monoxime, and 5 mM taurine) gassed with 100% O$_2$. Enzymatic digestion was initiated by adding collagenase type II (300 U/mL; Worthington) and hyaluronidase (0.1 mg/mL; Worthington) to the perfusion solution. When the heart became swollen after 45 min of digestion, the ventricles were removed, cut into several chunks, added to a BSA solution, and physically separated in a shaker (60–70 rpm for 5 min at 37˚C). The supernatants containing the fibroblasts were filtered through a cell strainer (100 μm pore size; BD Falcon) and gently centrifuged at 500 rpm for 3 min. The resultant supernatant was transferred into a new tube and centrifuged at 1,000 rpm for 10 min. FBs

were suspended and cultured in DMEM (HyClone) supplemented with penicillin, streptomycin, and 10% FBS (HyClone). Cells were ready to use for further experiments 48 hours after seeding. Trans-differentiation of FBs was induced by treatment with 10 ng/mL TGF-β (Pepprotech, 10035B) [17] or 100 nM AngII (Sigma-Aldrich, #4474-91-3) for 48 hours. Both TGF-β and AngII are known to be potent profiboric molecule. An IKK inhibitor, IKK VII, was obtained from Millipore (#401486) and used at a concentration of 1 μM for 48 hours.

## Conditioned media

COS-7 cells were cultured in DMEM (HyClone) supplemented with penicillin, streptomycin, and 10% FBS (HyClone). We synthesized a gene encoding CCN5 fused with a HA tag at its carboxy terminus and referred to it as CCN5-HA. 12 mL culture of COS-7 cells were transformed with 9 μg of pcDNA-CCN5-HA. The media were collected after 24 hours and referred to as CCN5-containing conditioned media (CM-CCN5). Control conditioned media (CM-Con) was similarly obtained but without the transformation with pcDNA-CCN5-HA. The concentration of CCN5-HA was typically 200~500 ng/mL when detected and calculated using anti-HA antibody (Roche, #11867423001). Blotting with an anti GAPDH antibody confirmed that the CM was not contaminated with cellular proteins.

## siRNA knockdown analysis

Cardiac FBs were transfected with 30 nM of siRNA targeting rat p53 (Bioneer, 1774203), 25 nM of On-TARGET plus Rat Smad7 siRNA (Dharmacon, L-093737-02), or a control siRNA (Bioneer, SN-1002) using DharmaFECT Transfection reagent (Dharmacon, T-2001-02). Cells were treated with CM-CCN5 or CM-Con 48 hr after transfection.

## Immunostaining

Cardiac FBs were seeded at 15,000 cells per well on 16 mm cover slips. When the cells reached 80% confluence, they were transfected with p53 siRNA, Smad7 siRNA, or a control siRNA. The cells were then treated with CM-CCN5 for 48 hours, fixed with 4% paraformaldehyde, permeabilized with 0.5% Triton X-100, and blocked with 5% BSA. Cells were incubated with an antibody against p53 (Cell Signaling Technology) or NFκB p65 (Santa Cruz), and were then incubated with secondary antibodies labelled with Alexa Fluor 488 or Alexa Fluor 594 (Invitrogen). Nuclei were stained with DAPI. Immunofluorescence was analyzed under a microscope equipped with a 60X objective lens and the appropriate filters (Olympus).

## Terminal transferase dUTP nick end-labeling assay (TUNEL)

Cardiac FBs were knocked-out with the p53 and Smad7 siRNAs or a control siRNA. After treatment with CM-CCN5 or CM-Con, cells on glass coverslips were stained using the Dead-End Fluorometric TUNEL kit (Promega). The presence of nicks in the DNA was confirmed using terminal deoxynucleotidyl transferase (TdT), an enzyme that catalyzes the addition of labeled dUTPs (6). TUNEL-positive cells were captured using a Fluoview FV 1000 confocal laser scanning microscope.

## Western blotting

RIPA buffer (50 mM Tris, pH 8.0, 150 mM NaCl, 0.1% SDS, and 1% Triton X-100) with Protease Inhibitor Cocktail Set III (Merck Millipore, 535140) was used to solubilize cells. The cell lysates were quantified using the Pierce BCA Protein Assay Kit (Thermo Scientific, 23227), and concentrations were normalized against bovine serum albumin. The proteins from the

quantified cell lysates were separated using sodium dodecyl sulfate polyacrylamide gel electrophoresis (SDS-PAGE) and were transferred to a polyvinylidene difluoride (PVDF) membrane (Merck Millipore, IPVH00010). The transferred blots were blocked with 5% non-fat skim milk and incubated with antibodies against α-SMA (Sigma-Aldrich, A5228), SMAD7 (Invitrogen), caspases (Antibody Sampler Kit, Cell Signaling), phospho–NFκB p65 (Ser 536) (Santa Cruz Biotechnology), NFκB p65 (Santa Cruz Biotechnology), cleaved PARP, PARP, phospho-IκBα, or IκBα (Cell Signaling Technology) overnight at 4˚C. After washing the blots with Tris buffered saline (TBS) with 0.1% Tween 20, the membranes were incubated with secondary antibodies conjugated with horseradish peroxidase (HRP) (Thermo Scientific, 31430) and washed again. The protein band signal was detected using a chemiluminescence solution (Dogen, DG-WP100). GAPDH was used as a normalizing control to validate the expression levels of the target proteins.

### Smad7 reporter gene assay

Rat MyoFBs were plated at $3 \times 10^5$ cells/well in 6-well culture plates and transfected with the Gluc-On Promoter reporter clone human Smad7 (GeneCopoeia; HMRM377850-PG04), pcDNA3.0-hCCN5, or an empty pcDNA3.0 vector using Lipofectamine 2000 (Invitrogen). After 48 hours, cell culture medium was collected, and luciferase assay signaling detection was performed using the Secrete-Pair Gaussia Luminescence Assay Kit (GeneCopoeia; LF061).

### Statistical analysis

Student's t-test and One-Way Analysis of Variance (ANOVA) were used for statistical analysis to determine the significance of the data. An asterisk (*P<0.05) or a double asterisk (**P<0.01), (***P<0.001) indicate significant probability.

## Results

### The p53 and NFκB signaling pathways are up-regulated in MyoFBs

Freshly prepared rat cardiac FBs were trans-differentiated to MyoFBs by treating them with the pro-fibrotic cytokine TGF-β for 48 hours. Western blotting of the cell extracts revealed that the protein level of p53 and the phosphorylation levels of p53 at serine 15 and serine 392 were elevated in MyoFBs compared to FBs. The protein and phosphorylation levels for NFκB p65 were also elevated in MyoFBs compared to FBs (Fig 1A). These data suggest that a pro-apoptotic (p53) and an anti-apoptotic (NFκB) signal are simultaneously elevated during trans-differentiation of FBs and that they remain balanced in MyoFBs. To test whether this phenomenon is specific to TGF-β–mediated differentiation, FB trans-differentiation was induced by the treatment with another pro-fibrotic cytokine angiotensin II (AngII). Western blotting revealed that similar patterns of up-regulation for both p53 and NFκB were induced in the trans-differentiated MyoFBs (Fig 1B). Freshly prepared FBs undergo trans-differentiation over the course of cell passaging even without any exogenous pro-fibrotic stimuli. Under our experimental conditions, we found that trans-differentiation of FBs was indeed induced over the course of passaging. The protein levels of p53 and NFκB concomitantly increased over the course of cell passaging (Fig 1C).

These data indicate that p53 and NFκB are simultaneously induced in MyoFBs compared to FBs, and this phenomenon is associated with trans-differentiation of FBs under various circumstances.

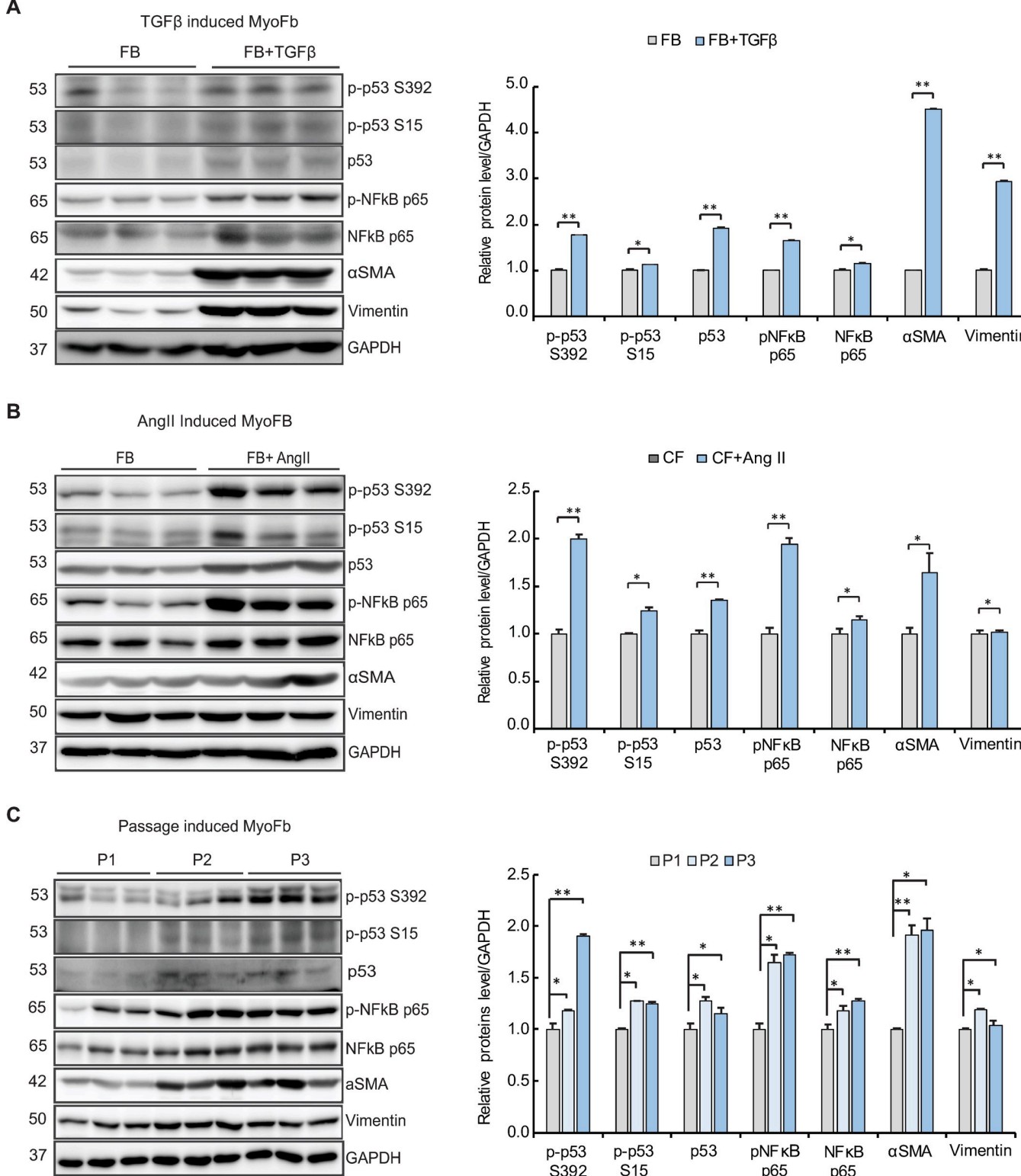

**Fig 1. The p53 and NFκB signaling pathways are up-regulated in MyoFBs.** Freshly prepared FBs were treated with (A) 10 ng/mL TGF-β or (B) 100 nM AngII, or (C) were subcultured. Cells lysates were then used for western blotting analysis. The expression levels of p53, phosphorylated p53 (p-p53 S15 or p-p53 S392), NFκB p65, phosphorylated NFκB p65 (p-NFκB p65), vimentin, and α-smooth muscle actin (α-SMA) were monitored by western blotting. GAPDH was used as a loading control. For each group, three independent cultures were prepared and treated (n = 3).

## p53 is involved in CCN5-mediated MyoFB apoptosis

To examine the role of p53 in MyoFBs, we utilized a siRNA that is designed to target p53 transcripts (p53 siRNA). MyoFBs were treated with either a control siRNA or the p53 siRNA, and were incubated in either control conditioned media (CM-Con) or CCN5-containing conditioned media (CM-CCN5) for 48 hours. Preparation of the conditioned media are described in the Materials and Methods section. Western blotting was then performed to examine the effects. Consistent with the data shown in Fig 1, the expression level of p53 increased and was further elevated when the MyoFBs were incubated in CM-CCN5. The protein levels of the executioner caspases, caspase 3 and caspase 7, and the levels of cleavage of these caspases were elevated after the cells were cultured in CM-CCN5. In addition, the cleavage of poly (ADP-ribose) polymerase (PARP-1) also increased in cells incubated in CM-CCN5. These data indicate that CCN5 induces apoptosis in MyoFBs, which is consistent with our previous results [14] (Fig 2A). Pre-incubation with the p53 siRNA blocked the expression of p53 in MyoFBs cultured either in CM-Con or CM-CCN5, and concomitantly blocked the expression and cleavage of caspase 3 and caspase 7 and the cleavage of PARP-1 (Fig 2A). TUNEL assay and immunostaining for p53 were performed simultaneously. In line with the western blotting results, incubation in CM-CCN5 induced the appearance of TUNEL-positive apoptotic cells, which was significantly blocked when the cells were pre-incubated with the p53 siRNA (Fig 2B).

These results indicate that CCN5 induces apoptosis in MyoFBs through the activity of p53, a pro-apoptotic protein.

## NFκB protects MyoFBs from apoptosis

We previously showed that when cells are treated with CCN5, NFκB is excluded from the nucleus [14], where it acts as an anti-apoptotic factor. Therefore, we examined the role of NFκB in MyoFBs using IKK VII, a selective inhibitor of IκB kinase. Treatment with IKK VII completely abolished the phosphorylation of IκBα and consequently diminished the phosphorylation level and the expression level of NFκB p65. This IKK VII-mediated inhibition of NFκB signaling was associated with activation of caspases 3 and 7 and cleavage of PARP-1 even when the MyoFBs were cultured in CM-Con (Fig 3A). TUNEL assay further confirmed that IKK VII induced MyoFB apoptosis without incubation in CM-CCN5. Co-treatment with IKK VII and CM-CCN5 did not synergistically enhance the level of apoptosis (Fig 3B).

These data suggest that NFκB plays a critical role in preventing apoptosis in MyoFBs and that CCN5 induces apoptosis through inhibition of the NFκB signaling pathway. These results and those from the previous section collectively suggest that a delicate balance between p53 and NFκB maintains MyoFB survival, and that CCN5 tips this balance over in favor of p53-induced apoptosis through inhibition of NFκB.

## CCN5 inhibits NFκB signaling through elevation of SMAD7

We previously observed that CCN5 inhibits the TGFβ-SMAD signaling pathway by elevating the expression level of SMAD7, an inhibitory SMAD [13, 14]. Interestingly, SMAD7 potentiates apoptosis in epithelial cells through inhibition of NFκB [18, 19]. Therefore, we hypothesized that SMAD7 is involved in the observed CCN5-mediated inhibition of NFκB. Western blotting analysis revealed that the expression level of SMAD7 robustly increased after cells were cultured in CM-CCN5. The levels of p53 and phosphorylated p53 increased, and the levels of phosphorylated IκB and NFκB p65 decreased. Pre-incubating the MyoFBs with a siRNA for SMAD7 (SMAD7 siRNA) completely blocked the CCN5-mediated increase in the level of SMAD7 and eliminated the effects of CCN5 on p53 and NFκB. In addition, pre-incubation

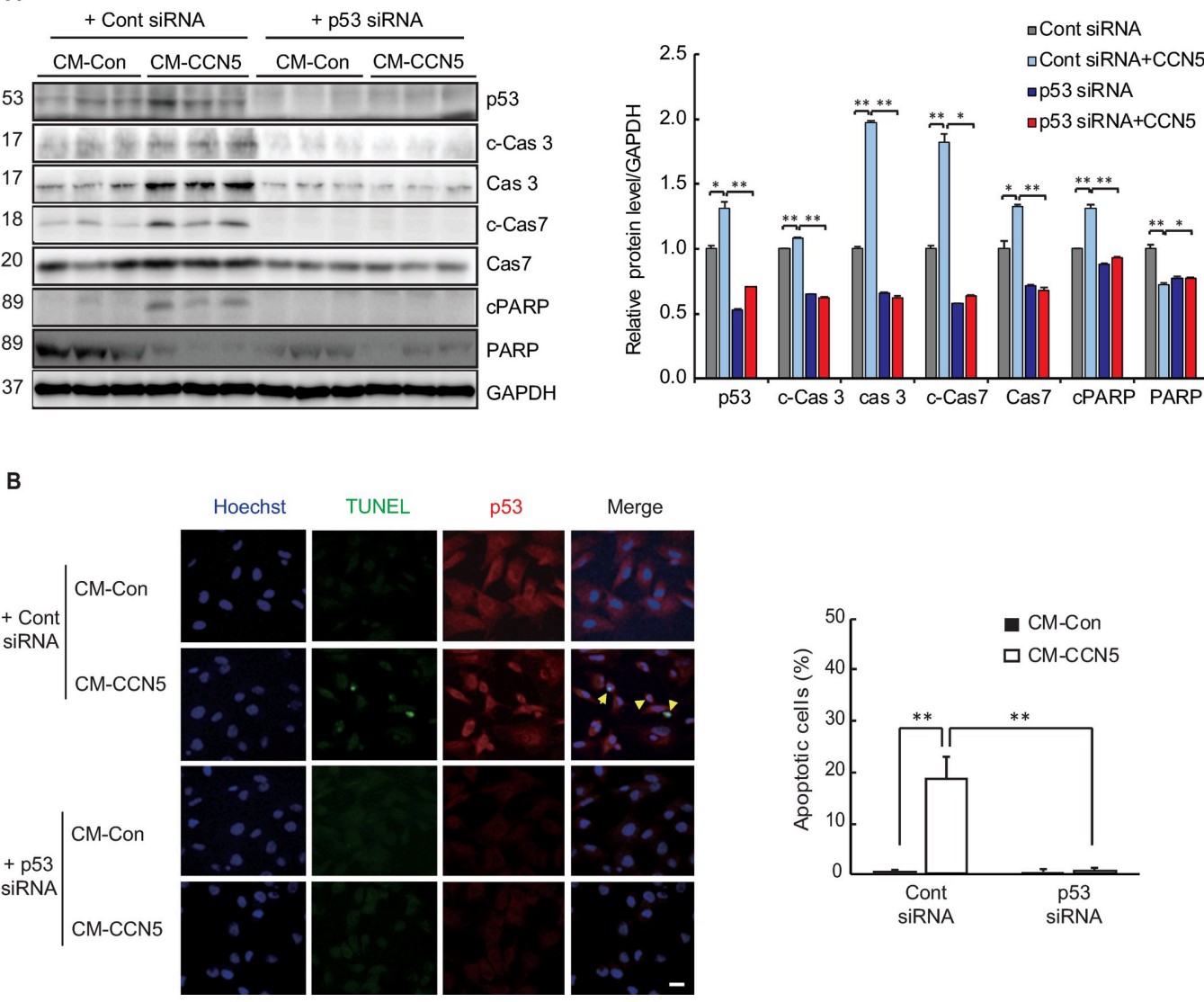

**Fig 2. p53 is involved in CCN5-mediated apoptosis of MyoFBs.** Freshly prepared FBs were transfected with 30 nM p53 siRNA and trans-differentiated to MyoFBs by treating them with 10 ng/mL TGF-β for 48 hours. MyoFBs were then cultured in control conditioned media (CM-Con) or CCN5-containing conditioned media (CM-CCN5) for a further 48 hours. (A) Cell lysates were then used for western blotting analysis of p53, caspase 3 (Cas3), cleaved caspase 3 (c-Cas3), caspase 7 (Cas7), cleaved caspase 7 (c-Cas7), Poly (ADP-ribose) polymerase (PARP), and cleaved PARP (cPARP). GAPDH was used as a loading control. (B) Cells were co-stained with TUNEL (green) and p53 (red). Hoechst dye was used for nuclear staining. TUNEL-positive apoptotic cells were counted and plotted. Scale bar is 50 μm. For each group, three independent cultures were prepared and treated (n = 3). *P<0.05, **P<0.01.

with the SMAD7 siRNA inhibited the cleavage of caspases 3 and 7 and PARP-1 (Fig 4A). TUNEL assay also revealed that pre-incubation with the SMAD7 siRNA inhibited CCN5-mediated apoptosis of MyoFBs, whereas the control siRNA exhibited no effects (Fig 4B).

These data suggest that SMAD7 plays a critical role in CCN5-mediated MyoFB apoptosis.

## CCN5 regulates the transcription of *SMAD7*

CCN5 acts as a transcriptional co-repressor and regulates the expression of TGFβ receptor II [16]. We hypothesized that CCN5 is involved in the transcriptional regulation of other genes, including *SMAD7*. Quantification of western blotting results showed that after incubating the

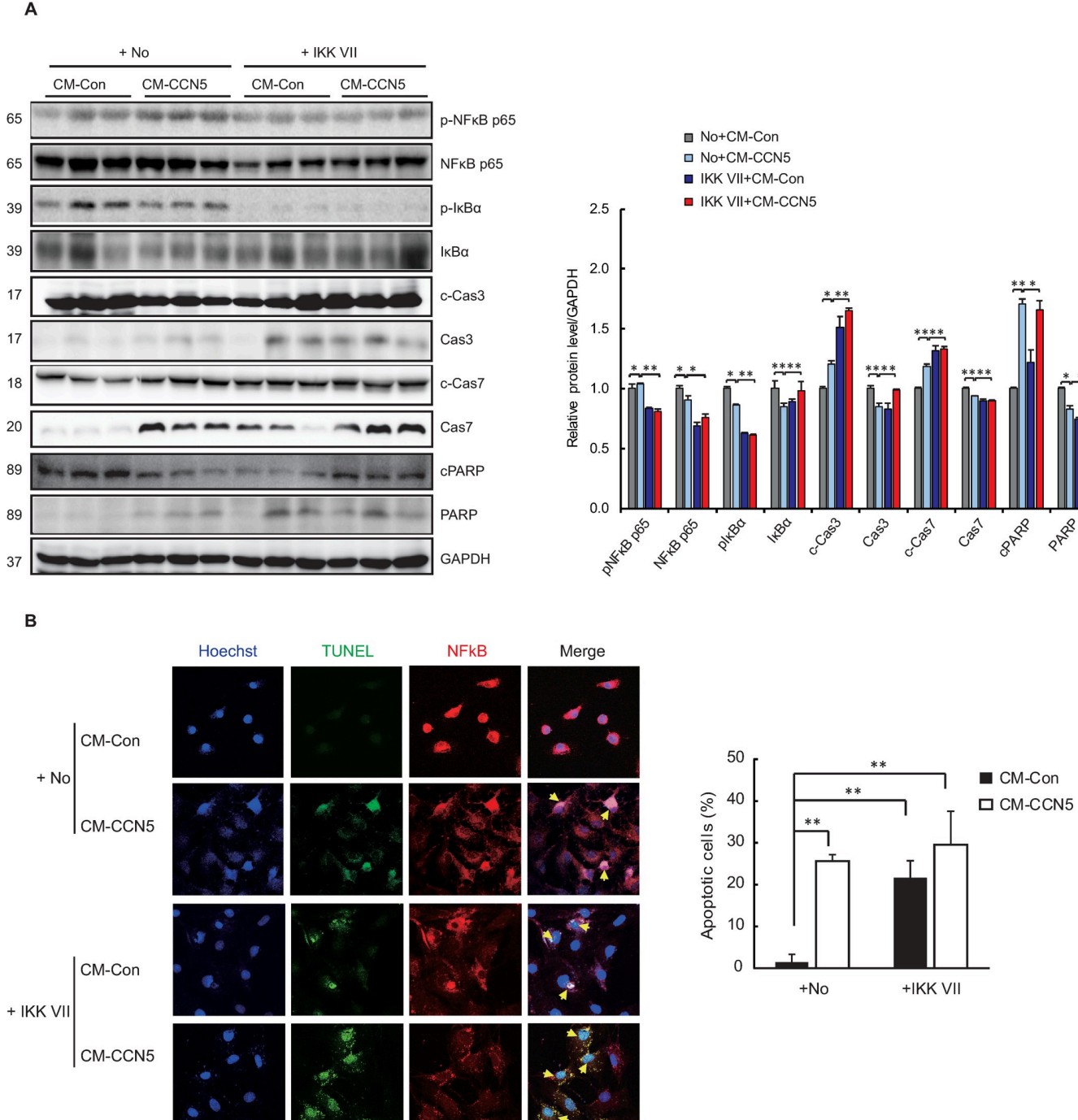

**Fig 3. NFκB protects MyoFBs from apoptosis.** Freshly prepared FBs were trans-differentiated to MyoFBs by treating them with 10 ng/mL TGF-β for 48 hours. MyoFBs were then cultured in control conditioned media (CM-Con) or CCN5-containing conditioned media (CM-CCN5) for a further 48 hours in the presence or absence of 1 μM IKK VII, a selective IκB kinase inhibitor. (A) Cell lysates were used for western blotting analysis of NFκB p65, phosphorylated NFκB p65 (p-NFκB p65), IκB, phosphorylated IκB (p-IκB), caspase 3 (Cas3), cleaved caspase 3 (c-Cas3), caspase 7 (Cas7), cleaved caspase 7 (c-Cas7), Poly (ADP-ribose) polymerase (PARP), and cleaved PARP (cPARP). GAPDH was used as a loading control. (B) Cells were co-stained with TUNEL (green) and p53 (red). Hoechst dye was used for nuclear staining. TUNEL-positive apoptotic cells were counted and plotted. Scale bar is 50 μm. For each group, three independent cultures were prepared and treated (n = 3). *P<0.05, **P<0.01.

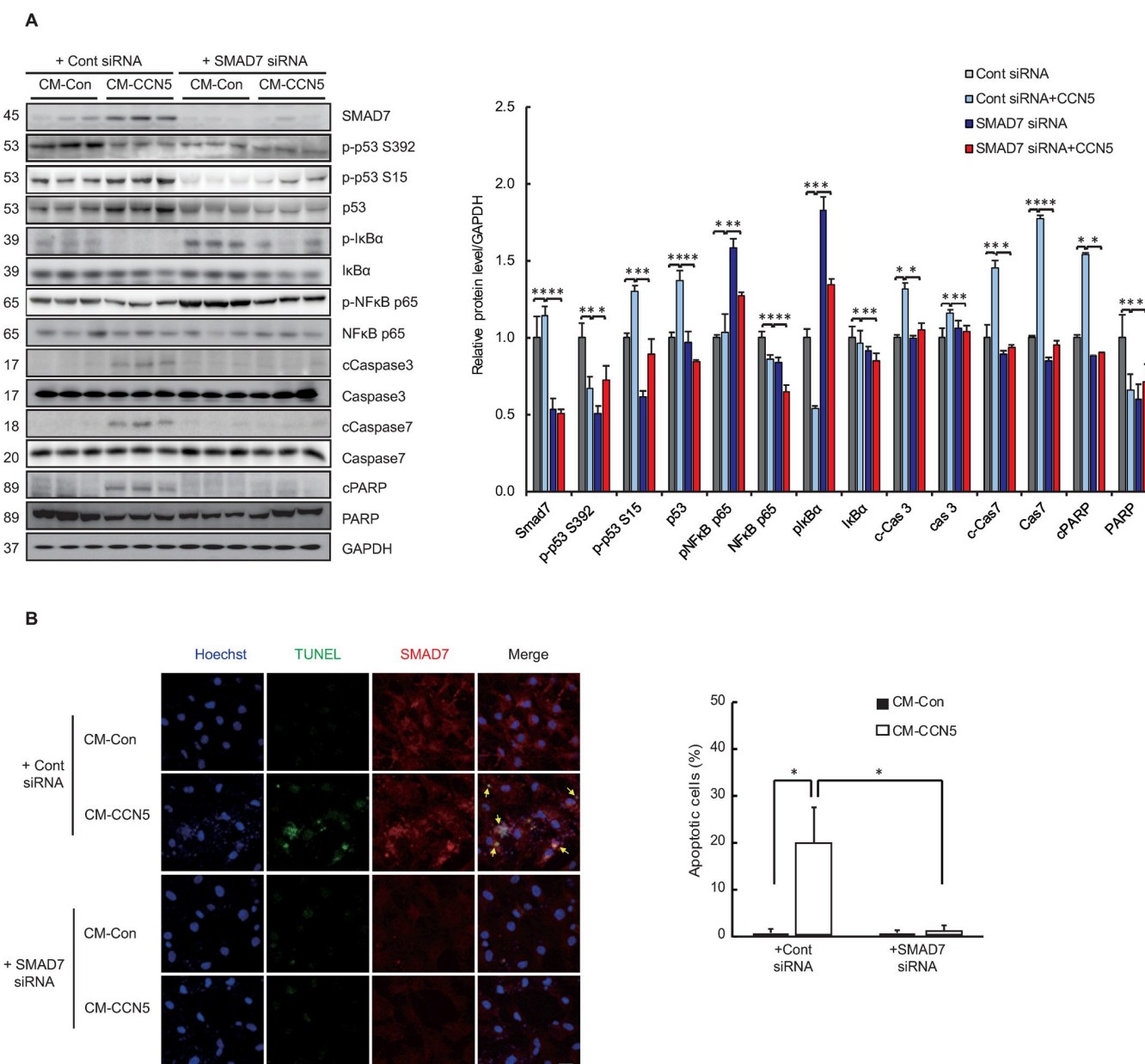

**Fig 4. CCN5 inhibits NFκB signaling through elevation of SMAD7 levels.** Freshly prepared FBs were transfected with 25 nM SMAD7 siRNA and were trans-differentiated to MyoFBs by treating them with 10 ng/mL TGF-β for 48 hours. MyoFBs were then cultured in control conditioned media (CM-Con) or CCN5-containing conditioned media (CM-CCN5) for a further 48 hours. (A) Cell lysates were used for western blotting analysis of SMAD7, p53, phosphorylated p53 (p-p53 S15 or p-p53 S392), NFκB p65, phosphorylated NFκB p65 (p-NFκB p65), IκB, phosphorylated IκB (p-IκB), caspase 3 (Cas3), cleaved caspase 3 (c-Cas3), caspase 7 (Cas7), cleaved caspase 7 (c-Cas7), Poly (ADP-ribose) polymerase (PARP), and cleaved PARP (cPARP). GAPDH was used as a loading control. (B) Cells were co-stained with TUNEL (green) and p53 (red). Hoechst dye was used for nuclear staining. TUNEL-positive apoptotic cells were counted and plotted. Scale bar is 50 μm. For each group, three independent cultures were prepared and treated (n = 3). *P<0.05, **P<0.01.

MyoFBs in CM-CCN5, the expression level of SMAD7 increased approximately 2.5-fold (Fig 5A). Quantitative RT-PCR also revealed that culturing cells in CM-CCN5 caused the level of *SMAD7* transcripts to increase approximately 3-fold (Fig 5B). We generated a reporter plasmid that drives the expression of secreted luciferase under the control of the *SMAD7* enhancer and promoter regions. This reporter plasmid was transformed into MyoFBs. Co-transformation

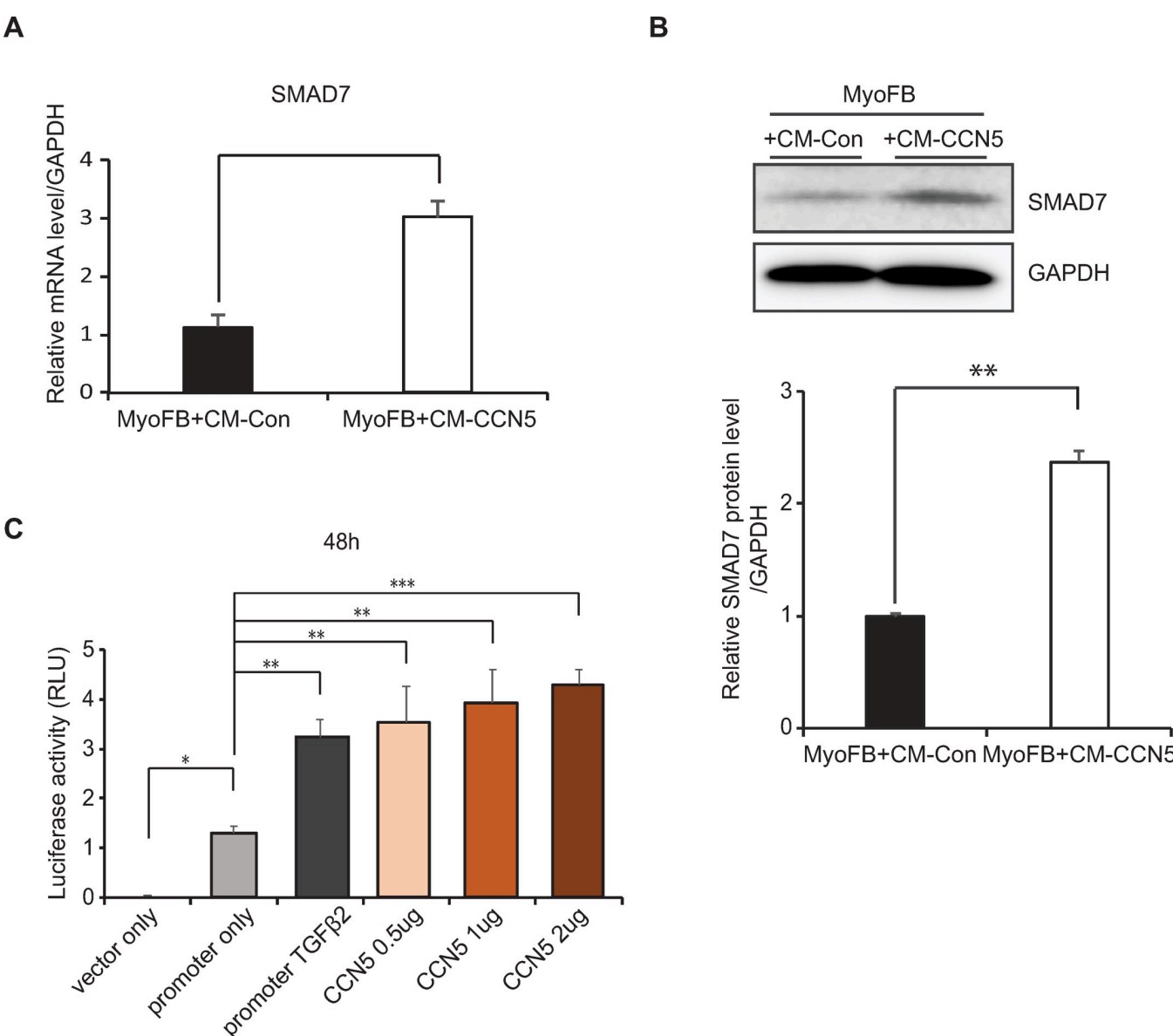

**Fig 5. CCN5 regulates the transcription of *SMAD7*.** Freshly prepared FBs were transfected with 25 nM SMAD7 siRNA and were trans-differentiated to MyoFBs by treating them with 10 ng/mL TGF-β for 48 hours. MyoFBs were then cultured in control conditioned media (CM-Con) or CCN5-containing conditioned media (CM-CCN5) for a further 48 hours. (A) Cell lysates were used for western blotting analysis of SMAD7. (B) Total RNA was prepared from cell lysates and used for qRT-PCR analysis. (C) A reporter construct for monitoring transcription from the *SMAD7* promoter was constructed. This reporter construct and CCN5-expressing plasmids were co-transformed into MyoFBs. Luciferase activity was measured 48 hours after transformation. The result is the ratio of luminescence intensities (RLU, Relative light Unit) of the Gluc over SEAP (S). Compare the normalized Gluc activity (RLU/S) of all samples. TGF-β was used as a positive control, and empty plasmid was used as a negative control. For each group, three independent cultures were prepared and treated (n = 3). *P<0.05, **P<0.01, ***p<0.001.

with CCN5-expressing plasmids significantly increased the activity of luciferase in the culture medium. Treatment with TGF-β, which served as a positive control, also increased the activity of luciferase to a similar extent.

These data suggest that CCN5 directly regulates the expression of *SMAD7* at the transcriptional level.

## Discussion

CF is associated with diverse cardiac conditions including myocardial infarction, hypertensive heart diseases, and diabetic cardiomyopathy. CF initially provides a protective mechanism with proper ECM protein deposition, which is beneficial for wound healing and tissue regeneration. However, when uncontrolled and sustained, CF causes excessive accumulation of ECM proteins and eventually leads to further cardiac complications such as diastolic dysfunction and arrhythmia [20–23]. Therefore, CF is one of the biggest concerns in the cardiovascular research community.

We previously found that the matricellular protein CCN5 inhibits CF through blocking the TGF-β–SMAD signaling pathway [13]. More strikingly, we found that CCN5 reverses pre-established CF through inducing apoptosis in MyoFBs but not in cardiomyocytes or FBs [14]. CCN5 accelerates the removal of MyoFBs through inducing the intrinsic pathway of apoptosis [14]. NFκB is involved in fibrosis in various organs [24–26] and it also possesses an anti-apoptotic activity in various contexts [27–30]. We previously showed that NFκB is involved in the regulation of trans-differentiation and apoptosis in MyoFBs [14]

In this study, we showed that NFκB and the pro-apoptotic protein p53 are up-regulated in MyoFBs that are generated *in vitro* in diverse ways (Fig 1). NFκB has dual roles in fibrosis, namely, (i) inducing fibrosis through activation of the TGF-β signaling pathway [24–26], and (ii) keeping MyoFBs viable though inhibition of apoptosis (this study). We showed that CCN5 induces p53-mediated apoptosis in MyoFBs (Fig 2) through negatively regulating NFκB (Fig 3). It is intriguing to note that a thin line of balance between pro-apoptotic and anti-apoptotic proteins is established in MyoFBs and that small changes in this balance can quickly lead to removal of MyoFBs. This phenomenon might reflect the transient nature of MyoFBs.

CCN5 consistently and robustly increases the expression level of SMAD7 in FBs and MyoFBs. SMAD7 potentiates the apoptosis of epithelial cells through inhibition of the NFκB signaling pathway [14]. In addition, SMAD7 sensitizes breast and gastric cancer cells to TNF-induced apoptosis through down-regulation of the NFκB signaling pathway [31–33]. Consistent with these reports, we showed that SMAD7 is involved in the down-regulation of NFκB in MyoFBs (Fig 4). On the basis of our findings, we conclude that CCN5 up-regulates SMAD7, which inhibits NFκB, to activate p53-mediated selective apoptosis of MyoFBs. CCN5 functions as a transcription cofactor that negatively regulates the expression of TGF-β receptor II [16]. In this study, we showed that CCN5 positively regulates the expression level of *SMAD7* at the transcriptional level (Fig 5). Therefore, CCN5 can act as a transcriptional activator or inhibitor depending on the context.

Targeted apoptosis of MyoFBs in other contexts has previously been reported. In the MyoFBs of scleroderma, an autoimmune disease characterized by multi-organ fibrosis, the pro-apoptotic BH3-only protein BIM and anti-apoptotic protein BCL-X$_L$ are kept in balance to ensure the survival of MyoFBs. The BH3 mimetic drug ABT-263 (navitoclax) prevents BCL-X$_L$ from checking BIM so that BIM can induce targeted apoptosis in MyoFBs [34, 35]. The anti-cancer drug elesclomol induces targeted apoptosis of MyoFBs in hypertrophic scars, although its underlying molecular mechanism of action is unclear [36]. Future studies should investigate whether BH3 mimetic drugs or elesclomol can induce selective apoptosis in MyoFBs derived from heart tissue.

Collectively, this study shows that survival of MyoFBs in the heart is dependent on a delicate balance between p53 and NFκB, and that CCN5 induces targeted apoptosis in MyoFBs through SMAD7-mediated inhibition of the NFκB signaling pathway.

## Supporting information

**S1 File.**
(PDF)

## Author Contributions

**Conceptualization:** Mai Tuyet Nguyen, Min-Ah Lee, Woo Jin Park.

**Data curation:** Mai Tuyet Nguyen, Min-Ah Lee, Woo Jin Park.

**Formal analysis:** Mai Tuyet Nguyen.

**Methodology:** Mai Tuyet Nguyen, Min-Ah Lee, Young-Kook Kim, Hyun Kook.

**Supervision:** Young-Kook Kim, Woo Jin Park.

**Validation:** Mai Tuyet Nguyen.

**Visualization:** Min-Ah Lee, Dongtak Jeong, Seung Pil Jang, Tae Hwan Kwak.

**Writing – original draft:** Mai Tuyet Nguyen, Woo Jin Park.

**Writing – review & editing:** Woo Jin Park.

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
