## [Decision Letter · Decision Letter 0]

6 Oct 2021

PONE-D-21-25986The matricellular protein CCN5 induces apoptosis in myofibroblasts through SMAD7-mediated inhibition of NFkBPLOS ONE

Dear Dr. Park,

Thank you for submitting your manuscript to PLOS ONE. After careful consideration, we feel that it has merit but does not fully meet PLOS ONE’s publication criteria as it currently stands. Therefore, we invite you to submit a revised version of the manuscript that addresses the points raised during the review process.

We look forward to receiving your revised manuscript.

Kind regards,

Dhyan Chandra, Ph.D.

Academic Editor

PLOS ONE

Journal Requirements:

“W.J.P. was supported by grants from National Research Foundation of Korea (2019R1A2C2085457, 2019R1A4A1028534) funded by the Korean Government (MSIP). Y.-K.K. and H. K. were supported by grants from National Research Foundation of Korea (2018R1A2B3001503, 2018R1A2B6001104, 2019R1A4A1028534, 2020R1C1C1006507). D.J. was supported by grants from National Research Foundation of Korea (2021R1A2C1008058, 2021R1A4A5032463)”

“W.J.P. was supported by grants from National Research Foundation of Korea (2019R1A2C2085457, 2019R1A4A1028534) funded by the Korean Government (MSIP). Y.- K.K. and H. K. were supported by grants from National Research Foundation of Korea (2018R1A2B3001503, 2018R1A2B6001104, 2019R1A4A1028534, 2020R1C1C1006507). D.J. was supported by grants from National Research Foundation of Korea (2021R1A2C1008058, 2021R1A4A5032463)”

We note that you have provided funding information within the Acknowledgements Section. Please note that funding information should not appear in the Acknowledgments section or other areas of your manuscript. We will only publish funding information present in the Funding Statement section of the online submission form.

“W.J.P. was supported by grants from National Research Foundation of Korea (2019R1A2C2085457, 2019R1A4A1028534) funded by the Korean Government (MSIP). Y.-K.K. and H. K. were supported by grants from National Research Foundation of Korea (2018R1A2B3001503, 2018R1A2B6001104, 2019R1A4A1028534, 2020R1C1C1006507). D.J. was supported by grants from National Research Foundation of Korea (2021R1A2C1008058, 2021R1A4A5032463)”

Reviewers' comments:

Reviewer's Responses to Questions

**Comments to the Author**

1. Is the manuscript technically sound, and do the data support the conclusions?

Reviewer #1: Yes

Reviewer #2: Yes

2. Has the statistical analysis been performed appropriately and rigorously? 

Reviewer #1: Yes

Reviewer #2: No

3. Have the authors made all data underlying the findings in their manuscript fully available?

Reviewer #1: Yes

Reviewer #2: Yes

4. Is the manuscript presented in an intelligible fashion and written in standard English?

Reviewer #1: Yes

Reviewer #2: Yes

5. Review Comments to the Author

Reviewer #1: In the submitted manuscript authors evaluated the mechanism of CCN5 mediated apoptosis of myofibroblasts involving p53 and NFkB. Further, SMAD7 was identified as the downstream target of CCN5 using luciferase reporter construct and reported to be positively regulated by CCN5 at transcriptional level. The manuscript is written with clarity and includes sufficient detail in each section. The experimental design is logical and most of the experiments were conducted in satisfactory manner. There are few minor things that authors should address:

1. In figure 1a loading control (GAPDH) seems to be high in FB+TGFβ samples. Authors are encouraged to quantify the western blots by densitometry to confirm if there is any overexpression of P53 and p-p53 (both S392 and S15).

2. Authors could have used high magnification images for Tunnel assay.

3. The statement in the last paragraph of the introduction “cardiac MyoFBs freshly isolated from rat hearts and trans-differentiated by treatment with TGF-�” is little confusing. It seems cardiac MyoFBs were transdifferentiated, though cardiac FBs were differentiated. Authors should review the statement.

Reviewer #2: The Manuscript "The matricellular protein CCN5 induces apoptosis in myofibroblasts through SMAD7-mediated inhibition of NFkB" by Nguyen et al is the extension of their earlier work where they have shown that the matricellular protein CCN5 reverses established cardiac fibrosis (CF) through inducing apoptosis in myofibroblasts (MyoFBs) but not in cardiomyocytes or fibroblasts (FBs). In this study, authors have the elucidate the molecular mechanisms underlying CCN5-mediated selective apoptosis of MyoFBs. Authors have demonstrated that the role of CCN5 in MyoFB-specific apoptosis using cardiac MyoFBs freshly isolated from rat hearts and trans-differentiated by treatment with TGF-β. They found that a balance between pro-apoptotic p53 and anti-apoptotic NFkB is established in MyoFBs and that CCN5 tips over the balance to favor p53, which results in apoptosis through SMAD7-mediated inhibition of NFkB. In the end, the authors have claimed that this work provides insight into the role of CCN5 in reversing CF and a basis for the development of anti-fibrotic therapies.

There are several concerns that need to be addressed.

General comments:

1. In the Abbreviations section, "E" of extracellular should be written in the capital later (line-47)

2. Extend the abbreviations "CCN5-5HA" (line- 114)

3. Animal Ethics Statement- should be written properly. It is written very haphazardly

Specific comments:

1. In the result section, as the authors mention in the materials and methods, they used GAPDH to validate the expression level of target proteins but they did not normalize the target protein with GAPDH (All western blot figures except fig.5)

2. The authors did not mention the molecular weight of any proteins (All figures)

3. In the result section, the authors use two type samples (FB and FB+ TGFβ2) but the western blot result showed the three bands of each group. If all three bands indicate the same group, then please mention it properly.

4. In Figure1A, the authors mention TGFβ2 but in Figure Legends and result section they mention only TGF-β. Please correct this.

5. What is the rationale of 30mM p53 siRNA dose selection because the authors did not explain the selection?

6. Again, the authors used 10ng/ml TGFβ and 100 nM AngII to transdifferentiate FB to MyoFBs but they did not explain why they used this concentration?

7. Similarly, they did not explain the dose selection of NFkB inhibitor (1µM)?

8. In the result section, sudden introduce that they used profibrotic proteins (TGFβ and AngII) to induced FB into MyoFB but they should be included in material and methods also.

9. In line-216, the authors should use words significantly blocked when the cells were pre-incubated with the p53 siRNA

10. Line-245, the authors used siRNA of SMAD7 but they did not mention the concentration, however, they mentioned the concentration (25mM SMAD7 siRNA) in the representative figure-4A

6. PLOS authors have the option to publish the peer review history of their article (what does this mean?). If published, this will include your full peer review and any attached files.

Reviewer #1: **Yes: **Ashish Kumar

Reviewer #2: No

---

## [Author Response · Author response to Decision Letter 0]

17 Nov 2021

Reviewer #1: In the submitted manuscript authors evaluated the mechanism of CCN5 mediated apoptosis of myofibroblasts involving p53 and NFkB. Further, SMAD7 was identified as the downstream target of CCN5 using luciferase reporter construct and reported to be positively regulated by CCN5 at transcriptional level. The manuscript is written with clarity and includes sufficient detail in each section. The experimental design is logical and most of the experiments were conducted in satisfactory manner. There are few minor things that authors should address:

1. In figure 1a loading control (GAPDH) seems to be high in FB+TGFβ samples. Authors are encouraged to quantify the western blots by densitometry to confirm if there is any overexpression of P53 and p-p53 (both S392 and S15).

 We have now quantitative graphs in figures 1~4.

2. Authors could have used high magnification images for Tunnel assay.

 We have now higher mag images with scale bars.

3. The statement in the last paragraph of the introduction “cardiac MyoFBs freshly isolated from rat hearts and trans-differentiated by treatment with TGF-�” is little confusing. It seems cardiac MyoFBs were transdifferentiated, though cardiac FBs were differentiated. Authors should review the statement.

 We revised the sentence to read " In this study, we attempted to further elucidate the role of CCN5 in MyoFB-specific apoptosis. FBs were freshly isolated from rat hearts, and MyoFBs were obtained by trans-differentiating FBs by treatment with TGF-�."

Reviewer #2: The Manuscript "The matricellular protein CCN5 induces apoptosis in myofibroblasts through SMAD7-mediated inhibition of NFkB" by Nguyen et al is the extension of their earlier work where they have shown that the matricellular protein CCN5 reverses established cardiac fibrosis (CF) through inducing apoptosis in myofibroblasts (MyoFBs) but not in cardiomyocytes or fibroblasts (FBs). In this study, authors have the elucidate the molecular mechanisms underlying CCN5-mediated selective apoptosis of MyoFBs. Authors have demonstrated that the role of CCN5 in MyoFB-specific apoptosis using cardiac MyoFBs freshly isolated from rat hearts and trans-differentiated by treatment with TGF-β. They found that a balance between pro-apoptotic p53 and anti-apoptotic NFkB is established in MyoFBs and that CCN5 tips over the balance to favor p53, which results in apoptosis through SMAD7-mediated inhibition of NFkB. In the end, the authors have claimed that this work provides insight into the role of CCN5 in reversing CF and a basis for the development of anti-fibrotic therapies.

There are several concerns that need to be addressed.

General comments:

1. In the Abbreviations section, "E" of extracellular should be written in the capital later (line-47)

 Corrected.

2. Extend the abbreviations "CCN5-5HA" (line- 114)

 We inserted a sentence that explains CCN5-HA in line 117~118, which read " We synthesized a gene encoding CCN5 fused with a HA tag at its carboxy terminus and referred to it as CCN5-HA."

3. Animal Ethics Statement- should be written properly. It is written very haphazardly

 We inserted a sentence in lines 93~95, which read " The investigation conforms to the Guide for the Care and Use of Laboratory Animals published by the US National Institutes of Health (NIH Publication No. 85–23, revised 1996)."

Specific comments:

1. In the result section, as the authors mention in the materials and methods, they used GAPDH to validate the expression level of target proteins but they did not normalize the target protein with GAPDH (All western blot figures except fig.5)

 We have now quantitative graphs in figures 1~4.

2. The authors did not mention the molecular weight of any proteins (All figures)

 All western blots now have molecular sizes marked.

3. In the result section, the authors use two type samples (FB and FB+ TGFβ2) but the western blot result showed the three bands of each group. If all three bands indicate the same group, then please mention it properly.

 For two groups (FB, FB+ TGFβ2), we had three independent cultures. Each lane in western blots represent samples of these three independent cultures. We added a sentence, " For each group, three independent cultures were prepared and treated (n=3)." at the ends of each figure legends.

4. In Figure1A, the authors mention TGFβ2 but in Figure Legends and result section they mention only TGF-β. Please correct this.

 In Figure 1A, TGF-β2 is now chaged to TGF-β.

5. What is the rationale of 30mM p53 siRNA dose selection because the authors did not explain the selection?

6. Again, the authors used 10ng/ml TGFβ and 100 nM AngII to transdifferentiate FB to MyoFBs but they did not explain why they used this concentration?

7. Similarly, they did not explain the dose selection of NFkB inhibitor (1µM)?

10. Line-245, the authors used siRNA of SMAD7 but they did not mention the concentration, however, they mentioned the concentration (25mM SMAD7 siRNA) in the representative figure-4A.

 For the issues 5,6,7,10, we now indicate the concentration and more detailed procedure in materials and methods section as well as figure legends. The indicated concentrations were those that were recommended by manufacturors or the best ones when tested in our lab.

8. In the result section, sudden introduce that they used profibrotic proteins (TGFβ and AngII) to induced FB into MyoFB but they should be included in material and methods also.

 This is now included in material and methods section.

9. In line-216, the authors should use words significantly blocked when the cells were pre-incubated with the p53 siRNA

 We added a word "significantly" in line 219.

---

## [Editor Report · Decision Letter 1]

5 Apr 2022

PONE-D-21-25986R1The matricellular protein CCN5 induces apoptosis in myofibroblasts through SMAD7-mediated inhibition of NFkBPLOS ONE

Dear Dr. Park,

Thank you for submitting your manuscript to PLOS ONE. After careful consideration, we feel that it has merit but does not fully meet PLOS ONE’s publication criteria as it currently stands. Therefore, we invite you to submit a revised version of the manuscript that addresses the points raised during the review process.

We look forward to receiving your revised manuscript.

Kind regards,

Dhyan Chandra, Ph.D.

Academic Editor

PLOS ONE
---

## [Author Response · Author response to Decision Letter 1]

29 Apr 2022

Reviewer #1: In the submitted manuscript authors evaluated the mechanism of CCN5 mediated apoptosis of myofibroblasts involving p53 and NFκB. Further, SMAD7 was identified as the downstream target of CCN5 using luciferase reporter construct and reported to be positively regulated by CCN5 at transcriptional level. The manuscript is written with clarity and includes sufficient detail in each section. The experimental design is logical and most of the experiments were conducted in satisfactory manner. There are few minor things that authors should address:

1. In figure 1a loading control (GAPDH) seems to be high in FB+TGFβ samples. Authors are encouraged to quantify the western blots by densitometry to confirm if there is any overexpression of P53 and p-p53 (both S392 and S15).

 We have now quantitative graphs in figures 1~4.

2. Authors could have used high magnification images for Tunnel assay.

 We have now higher mag images with scale bars.

3. The statement in the last paragraph of the introduction “cardiac MyoFBs freshly isolated from rat hearts and trans-differentiated by treatment with TGF-β” is little confusing. It seems cardiac MyoFBs were transdifferentiated, though cardiac FBs were differentiated. Authors should review the statement.

 We revised the sentence to read " In this study, we attempted to further elucidate the role of CCN5 in MyoFB-specific apoptosis. FBs were freshly isolated from rat hearts, and MyoFBs were obtained by trans-differentiating FBs by treatment with TGF-β."

Reviewer #2: The Manuscript "The matricellular protein CCN5 induces apoptosis in myofibroblasts through SMAD7-mediated inhibition of NFκB" by Nguyen et al is the extension of their earlier work where they have shown that the matricellular protein CCN5 reverses established cardiac fibrosis (CF) through inducing apoptosis in myofibroblasts (MyoFBs) but not in cardiomyocytes or fibroblasts (FBs). In this study, authors have the elucidate the molecular mechanisms underlying CCN5-mediated selective apoptosis of MyoFBs. Authors have demonstrated that the role of CCN5 in MyoFB-specific apoptosis using cardiac MyoFBs freshly isolated from rat hearts and trans-differentiated by treatment with TGF-β. They found that a balance between pro-apoptotic p53 and anti-apoptotic NFκB is established in MyoFBs and that CCN5 tips over the balance to favor p53, which results in apoptosis through SMAD7-mediated inhibition of NFκB. In the end, the authors have claimed that this work provides insight into the role of CCN5 in reversing CF and a basis for the development of anti-fibrotic therapies.

There are several concerns that need to be addressed.

General comments:

1. In the Abbreviations section, "E" of extracellular should be written in the capital later (line-47)

 Corrected. (Line 50)

2. Extend the abbreviations "CCN5-5HA" (line- 114)

 We inserted a sentence that explains CCN5-HA in line 121~122, which read " We synthesized a gene encoding CCN5 fused with a HA tag at its carboxy terminus and referred to it as CCN5-HA."

3. Animal Ethics Statement- should be written properly. It is written very haphazardly

 We inserted a sentence in lines 93~95, which read " The investigation conforms to the Guide for the Care and Use of Laboratory Animals published by the US National Institutes of Health (NIH Publication No. 85–23, revised 1996)."

Specific comments:

1. In the result section, as the authors mention in the materials and methods, they used GAPDH to validate the expression level of target proteins but they did not normalize the target protein with GAPDH (All western blot figures except fig.5)

 We have now quantitative graphs in figures 1~4.

2. The authors did not mention the molecular weight of any proteins (All figures)

 All western blots now have molecular sizes marked.

3. In the result section, the authors use two type samples (FB and FB+ TGFβ2) but the western blot result showed the three bands of each group. If all three bands indicate the same group, then please mention it properly.

 For two groups (FB, FB+ TGFβ2), we had three independent cultures. Each lane in western blots represent samples of these three independent cultures. We added a sentence, " For each group, three independent cultures were prepared and treated (n=3)." at the ends of each figure legends.

4. In Figure1A, the authors mention TGFβ2 but in Figure Legends and result section they mention only TGF-β. Please correct this.

 In Figure 1A, TGF-β2 is now changed to TGF-β.

5. What is the rationale of 30mM p53 siRNA dose selection because the authors did not explain the selection?

6. Again, the authors used 10ng/ml TGFβ and 100 nM AngII to transdifferentiate FB to MyoFBs but they did not explain why they used this concentration?

7. Similarly, they did not explain the dose selection of NFkB inhibitor (1µM)?

10. Line-245, the authors used siRNA of SMAD7 but they did not mention the concentration, however, they mentioned the concentration (25mM SMAD7 siRNA) in the representative figure-4A.

 For the issues 5,6,7,10, we now indicate the concentration and more detailed procedure in materials and methods section as well as figure legends. The indicated concentrations were those that were recommended by manufacturers or the best ones when tested in our lab.

8. In the result section, sudden introduce that they used profibrotic proteins (TGFβ and AngII) to induced FB into MyoFB but they should be included in material and methods also.

 This is now included in material and methods section.

9. In line-216, the authors should use words significantly blocked when the cells were pre-incubated with the p53 siRNA

 We added a word "significantly" in line 233.

---

## [Decision Letter · Decision Letter 2]

27 May 2022

The matricellular protein CCN5 induces apoptosis in myofibroblasts through SMAD7-mediated inhibition of NFkB

PONE-D-21-25986R2

Dear Dr. Park,

We’re pleased to inform you that your manuscript has been judged scientifically suitable for publication and will be formally accepted for publication once it meets all outstanding technical requirements.

Kind regards,

Dhyan Chandra, Ph.D.

Academic Editor

PLOS ONE

Additional Editor Comments (optional):

Reviewers' comments:

Reviewer's Responses to Questions

**Comments to the Author**

1. If the authors have adequately addressed your comments raised in a previous round of review and you feel that this manuscript is now acceptable for publication, you may indicate that here to bypass the “Comments to the Author” section, enter your conflict of interest statement in the “Confidential to Editor” section, and submit your "Accept" recommendation.

Reviewer #1: (No Response)

Reviewer #2: All comments have been addressed

2. Is the manuscript technically sound, and do the data support the conclusions?

Reviewer #1: Yes

Reviewer #2: Yes

3. Has the statistical analysis been performed appropriately and rigorously? 

Reviewer #1: Yes

Reviewer #2: Yes

4. Have the authors made all data underlying the findings in their manuscript fully available?

Reviewer #1: Yes

Reviewer #2: Yes

5. Is the manuscript presented in an intelligible fashion and written in standard English?

Reviewer #1: Yes

Reviewer #2: Yes

6. Review Comments to the Author

Reviewer #1: (No Response)

Reviewer #2: (No Response)

7. PLOS authors have the option to publish the peer review history of their article (what does this mean?). If published, this will include your full peer review and any attached files.

Reviewer #1: No

Reviewer #2: No

---

## [Editor Report · Acceptance letter]

22 Jul 2022

PONE-D-21-25986R2 

The matricellular protein CCN5 induces apoptosis in myofibroblasts through SMAD7-mediated inhibition of NFkB 

Dear Dr. Park:

I'm pleased to inform you that your manuscript has been deemed suitable for publication in PLOS ONE. Congratulations! Your manuscript is now with our production department. 

Kind regards, 

on behalf of

Dr. Dhyan Chandra 

Academic Editor

PLOS ONE